# Cellular and Molecular Alterations Underlying Abnormal Bone Growth in X-Linked Hypophosphatemia

**DOI:** 10.3390/ijms23020934

**Published:** 2022-01-15

**Authors:** Rocío Fuente, María García-Bengoa, Ángela Fernández-Iglesias, Helena Gil-Peña, Fernando Santos, José Manuel López

**Affiliations:** 1Division of Pediatrics, Department of Medicine, Faculty of Medicine, University of Oviedo, 33006 Oviedo, Spain; fuenterocio@uniovi.es (R.F.); mariagbengoa18@gmail.com (M.G.-B.); angelafiglesias@gmail.com (Á.F.-I.); hgilpena@gmail.com (H.G.-P.); fsantos@uniovi.es (F.S.); 2Instituto de Investigación Sanitaria del Principado de Asturias (ISPA), 33011 Oviedo, Spain; 3Institute of Physiology, Center for Integrative Human Physiology (ZIHP), University of Zurich, Winterthurerstrasse 190, CH-8057 Zurich, Switzerland; 4Research Center for Emerging Infections and Zoonoses (RIZ), University of Veterinary Medicine Hannover, 30559 Hanover, Germany; 5Department of Pediatrics, Hospital Universitario Central de Asturias (HUCA), 33011 Oviedo, Spain; 6Department of Morphology and Cellular Biology, Faculty of Medicine, University of Oviedo, 33006 Oviedo, Spain

**Keywords:** X-linked hypophosphatemia, XLH, FGF23, PHEX, phex, growth plate, GP, Hyp mice

## Abstract

X-linked hypophosphatemia (XLH), the most common form of hereditary hypophosphatemic rickets, is caused by inactivating mutations of the phosphate-regulating endopeptidase gene (PHEX). XLH is mainly characterized by short stature, bone deformities and rickets, while in hypophosphatemia, normal or low vitamin D levels and low renal phosphate reabsorption are the principal biochemical aspects. The cause of growth impairment in patients with XLH is not completely understood yet, thus making the study of the growth plate (GP) alterations necessary. New treatment strategies targeting FGF23 have shown promising results in normalizing the growth velocity and improving the skeletal effects of XLH patients. However, further studies are necessary to evaluate how this treatment affects the GP as well as its long-term effects and the impact on adult height.

## 1. Introduction

Defining the molecular and cellular mechanisms underlying bone development is essential for getting a better understanding of human skeletal diseases [1]. Endochondral ossification is the process by which most of the bones of the body are formed from cartilage in early fetal development, and it continues throughout the period of growth. Longitudinal growth occurs within the long bones at the growth plate (GP). During childhood, the GP forms cartilage by the proliferation and hypertrophy of chondrocytes and synthesis of a specific extracellular matrix. Cartilage is subsequently calcified, degraded, and replaced by osseous tissue in a tightly regulated manner, allowing adequate growth of the body. X-linked hypophosphatemia (XLH) is the most frequent inherited cause of hypophosphatemic rickets and osteomalacia. In this disease, hypomineralization leads to severe impaired growth and skeletal deformities. The bone growth is the result of the GP activity that involves chondrocyte proliferation, hypertrophy/differentiation, apoptosis, cartilage matrix synthesis and remodeling of the cartilage into bone. This process is tightly regulated by a complex interaction of molecular signals acting systemically and locally within the GP. Disturbances of the skeletal growth are associated with alterations in the activity/regulation of the GP cartilage. Such disturbances are little known in XLH and will be the basis of the present review.

## 2. X-Linked Hypophosphatemia (XLH)

XLH (OMIM #307800) is the most common inherited form of rickets, with an estimated frequency of 1 in 20,000 patients [2,3,4,5]. Other common names for the disease are: X-linked vitamin D-resistant rickets, hypophosphatemic vitamin D-resistant rickets, X-linked hypophosphatemic rickets, X-linked dominant hypophosphatemic rickets, X-linked rickets, familial hypophosphatemic rickets, familial hypophosphatemia, hypophosphatemic rickets, familial hypophosphatemic rickets and genetic rickets.

XLH is an inherited disease of phosphate metabolism caused by inactivating mutations of the phosphate-regulating endopeptidase homolog X-Linked (PHEX) gene [4,6,7]. More than 300 mutations of the PHEX gene have been reported in patients with XLH [4,8]. XLH follows a dominant inheritance linked to the X chromosome and causes growth retardation, rickets, osteomalacia, spontaneous dental abscesses, hearing difficulties, enthesopathy, early osteoarthritis (OA), muscular dysfunction, bone deformations and bone pain [9,10,11,12] (Table 1).

### 2.1. Pathogenesis of XLH

The genetic basis for XLH is the loss of function of the PHEX gene, located in chromosome Xp22; this loss results in an elevation of the serum levels of fibroblast growth factor 23 (FGF23), and impaired renal production of 1,25-dihydroxyvitamin D (1,25-(OH)_2_D). FGF23 has been reported to be responsible for hypophosphatemia and reduced 1,25-(OH)_2_D levels in animal models [13,14]. Although no direct link has been demonstrated between PHEX and FGF23 [9], a relationship between PHEX inactivation and the over-activation of M13 family members has been described [15,16,17,18].

PHEX is a member of the M13 family of type II cell surface zinc-dependent proteases, a group of neutral endopeptidases involved in the proteolytic processing of extracellular matrix proteins, and is predominately expressed in osteoblasts and osteocytes in bone; however, it is also expressed in the lung, brain, ovary, testicle, and muscle. The PHEX human gene shows an identity relation of 96% with that of a mouse [2,11,19].

FGF23 is regulated by 1,25-(OH)_2_D, parathyroid hormone (PTH), calcium, and local bone-derived factors, and is mainly produced by osteoblasts and osteocytes in response to increased extracellular phosphate and circulating 1,25-(OH)_2_D or calcitriol [20]. FGF23 acts by inhibiting the expression of the sodium phosphate (NaPi) co-transporters—NaPi-IIa and NaPi-IIc—at the renal proximal tubule, and inhibiting the synthesis of calcitriol, leading to a reduction in both renal and digestive phosphate reabsorption and renal phosphate wasting. It also causes downregulation of the renal 1α-hydroxylase enzyme (CYP27B1) and upregulation of the 24-hydroxylase enzyme (CYP24A1), leading to impaired 1.25(OH)_2_D synthesis and increased degradation. This dual defect in phosphate metabolism gives rise to poor bone mineralization and fractures [18,21,22].

Studies performed in mouse models with FGF receptor deletions demonstrated that fibroblast growth factor receptor 1 (FGFR1) is the principal mediator of FGF23-induced phosphaturia with FGFR3/4 having a minor role. In this regard, a close relationship between FGF23 and α-Klotho, a single-pass transmembrane protein, has been described, as α-Klotho acts as a co-receptor of FGFR1, increasing the affinity of the ligand–receptor union. This suggests that FGF23 requires α-Klotho for its activity, making α-Klotho another critical regulator of phosphate homeostasis and, directly and/or indirectly, of bone mineralization. It has been also described that the ablation of FGFR1 in bone partially rescues the excessive FGF23 secretion [12,14,23].

Bone disorders in which FGF23 shows elevated levels in the presence of normal PHEX expression have been described. Table 2 shows the main hypophosphatemic syndromes of genetic origin related to an elevation on FGF23 levels.

### 2.2. Hyp Mice

Mouse models have been extremely useful for getting a better understanding of the molecular mechanisms regulating chondrocyte formation, maturation and differentiation [1]. In this regard, although several animal models have been described, the Hyp mouse discovered in 1976 by Eva M. Eicher [24] is the most well-known. The Hyp mouse has a spontaneous deletion in the PHEX gene that begins in exon 16 extending through 3′-UTR. Hyp is the most completely characterized model of human XLH and reproduces the symptoms of the disease, including rickets, kyphosis of the thoracic vertebrae, prominent bowing of the femur, and dental abnormalities. A sexual dimorphism has been also reported, since osseous manifestations are more severe and uniformly presented in male mice [4].

Hyp mice have reduced body-weight, a shortened tail, rickets and histological features of osteomalacia, as well as enthesopathies [12]. Biochemically, Hyp mice are characterized by hypophosphatemia, hyperphosphaturia, elevated circulating levels of FGF23, and suppression of the renal production of 1.25(OH)2D. Low proximal tubular reabsorption of phosphate due to defective renal transport in the proximal renal tube is also a main characteristic, which has been demonstrated by a depressed expression of the NaPi-IIa and NaPi-IIc co-transporters in brush border membranes of the proximal nephron in Hyp mice [4]. Interestingly, Hyp females are fertile and can raise their young, but not all Hyp males can sire offspring [25].

## 3. XLH and Growth

XLH patients show wide phenotypic variability [4,26], but short stature and body disproportion are key clinical manifestations. Adults with XLH have significantly lower final height, up to 20 cm less, on average, than normal reference values and the mean standard deviation score (SDS), standing at around −1.9. XLH in adults is also associated with increased risk of pseudofractures, osteoarthritis, enthesopathy, and a variety of musculoskeletal symptoms, including pain and stiffness that decrease mobility and impact daily function [27,28]. Growth reduction is disharmonic with the lower body segment being relatively shorter than the upper body segment, so that the adult leg length is around -2.7 SDS. Classically, clinical manifestations of the disease are more severe in males than in females [12], although this has not been confirmed in all series of patients. A multicenter German study on 127 children with untreated XLH found that in children older than 5 years of age, stunting was more pronounced in boys than in girls, with the height SDS being —3.0 ± 1.0 vs. —2.1 ± 1.3, respectively; conversely, in children under 1 year of age, no differences were described [29,30].

XLH can be considered as a cartilage disorder, with a very high morbidity, that affects growth and generates pain, thus greatly reducing the quality of life of patients [31]. The pathogenic basis of growth impairment and cartilage disorder in XLH is not well understood and the majority of studies are focused on the phosphorus metabolism. Now we know that the severity of growth impairment is not related to the degree of hypophosphatemia, and that it is not directly dependent on the intensity of bone deformities.

Animal studies have been used to investigate the pathogenesis of growth and bone-cartilage impairment in XLH [12,32,33]. The influence of gender and gene dosage on the growth of Hyp mice was explored by Beck-Nielsen et al. (2010), who determined that while hemizygous PHEX−/Y mice displayed comparable skeletal abnormalities to heterozygous PHEX−/PHEX+ mice, homozygous female PHEX−/PHEX− mice exhibited more marked abnormalities, despite having equivalent serum phosphate and FGF23 concentrations [12]. Research studies on cartilage alterations in XLH are necessary to analyze the mechanism underlying growth impairment in XLH.

The control of longitudinal bone growth is performed by the GP cartilage, so any growth retardation and/or abnormal bone development is associated with a functional/structural alteration of this specialized tissue. Chondrocytes within the GP elongate the bone by proliferation, hypertrophy and synthesis of the extracellular matrix. The continual production of cartilage is coordinated within a process of cartilage replacement by bone, in which osteoblasts promote the death of chondrocytes and occupy the space they leave. In this way, there is a balance between GP activity and mineralization [1,34,35,36]. Yang et al. demonstrated that hypertrophic chondrocytes cannot only die, but they can also differentiate and become osteoblast cells [37]. The contribution of chondrocytes to the osteoblast lineage has potential implications in bone development, disease and repairment, although it is not yet well-characterized.

GP extends from the epiphyseal bone to the metaphyseal bone in children, and is constituted by hyaline cartilage, which is mainly formed of matrix and chondrocytes organized in columns [29]. However, in the postnatal GP, there are four layers of chondrocytes that contribute to the longitudinal growth of long bones (Figure 1). At the epiphyseal end, the reserve zone contains the resting chondrocytes, key for the maintenance of the GP integrity. As cells within the resting zone begin to divide, the proliferating zone is formed. Proliferative chondrocytes differentiate into pre-hypertrophic and, subsequently, hypertrophic chondrocytes. Hypertrophy of chondrocytes is characterized by a widespread increase in the volume of cells, a process divided into two phases of true hypertrophy (phase 1 and 3) separated by a phase of swelling (phase 2). Phase 3 of volume enlargement has been described to be regulated by local IGF1, suggesting that it may play an important role in the establishment of cell size [1,38,39,40,41]. The rate of bone elongation depends on the velocity and magnitude of this chondrocyte volume enlargement, and the size of the terminal hypertrophic chondrocyte is the main factor determining the final height of an individual [29,41,42]. To better understand skeletal growth and cartilage disorders, such as XLH, further studies of the chondrocyte differentiation process are required.

## 4. Growth Plate Alterations in XLH

Even though growth retardation is a major clinical manifestation of XLH, little is known about the underlying GP alterations in this disease. The study of the GP is extremely difficult in the clinical setting and justifies the use of animal models to better understand the mechanisms interfering with normal growth. Animal studies have shown profound abnormalities in the GP regarding the general organization, differentiation, and regulation. Better understanding of the GP structure and dynamics will provide the basis to explain growth impairment and metaphyseal deformities in XLH.

### 4.1. Changes in the Organization of GP

XLH’s GP is very irregular, presenting abnormal orientation of distal chondrocytes, aberrant vascularization of chondro-osseous tissue, and irregular mineralization, which might explain the abnormal trabecular organization at the primary spongiosa. A Hyp’s GP width is higher than in normal mice [6]. Different publications have showed that the expanded Hyp GP may be mainly caused by an expansion of the hypertrophic chondrocyte layer due to a diminished percentage of terminal chondrocytes in cellular apoptosis [4,16], while others observed impaired chondrocyte proliferation and a decreased proliferative chondrocyte zone [6]. GP chondrocytes are organized in columns distributed in parallel in normal individuals, while XLH patients show unpolarized and irregularly organized chondrocytes (Figure 2). An outstanding result of morphological analysis in XLH animal models showed that the chondrocyte columns from Hyp GP are not uniformly altered, but certain modifications affect some columns more than others. This finding may have great pathophysiological relevance and is very different from that observed in other diseases such as uremia, in which the GP is deeply affected, but in a more homogeneous way. It is remarkable that different chondrocytes can respond so differently despite the fact that they are all part of the same tissue and are exposed to a common metabolic environment, characterized in the Hyp mouse by hypophosphatemia, with a relative deficit of active vitamin D and an excess of FGF23.

### 4.2. Impaired Hypertrophic Chondrocyte Differentiation

Hypertrophic chondrocytes are specialized cells considered to be the end state of the chondrocyte differentiation pathway, and are essential for bone growth. They are characterized by expression of type X collagen encoded by the Col10a1 gene, synthesis of a calcified cartilage matrix, and a widespread increase in cell volume [41]. XLH’s hypertrophic chondrocytes present a different phenotype and become progressively flat as they move towards the end of the GP, even losing their polarity (Figure 2). The BrdU signal can be seen in some, but not all, hypertrophic chondrocytes (Figure 2, black star), suggesting that they could be re-entering the cell cycle and undergoing cell division, instead of entering an apoptotic process [5] or differentiating into bone-forming cells or osteoblasts. Liu Yang et al. demonstrated in their study [37] that normal terminal differentiated hypertrophic chondrocytes do not only die, but they can have two fates: they can either die leaving the space for new bone formation or become osteoblasts. This discovery of a chondrocyte-to-osteoblast lineage implies a revision of the concept of the ontogeny of osteoblasts, with clear implications for the control of bone homeostasis and the interpretation of the underlying pathological bases in XLH, which is completely unexplored.

Chondrocyte hypertrophy is an essential contributor to longitudinal bone growth, but recent data suggest that these cells also play fundamental roles in signaling to other skeletal cells (as osteoclasts and osteoblasts), thus coordinating endochondral ossification [43,44,45]. Therefore, a better understanding of the processes that control chondrocyte hypertrophy in the GP, as well as in other cartilage tissues, is required for improved management of both XLH and other skeletal growth and cartilage disorders.

### 4.3. Alteration of Growth Factors and Local Regulators Associated with Growth Plate

The PHEX protein was localized in the proliferating and hypertrophic chondrocytes of long bone GP and found to be absent in the GP chondrocytes of Hyp mice with mutated PHEX [46]. This finding could partly prove that the abnormal PHEX function may contribute directly to the cartilage abnormalities typically found in Hyp mice. A recent report by Lui et al. [43,47] demonstrated that SOX9, a master regulatory transcription factor for chondrogenesis, enhanced PHEX promoter activity in C5.18 chondrocytes in vitro. However, in vivo, SOX9 is believed to predominantly enhance the proliferating chondrocyte population as expression decreases in hypertrophic and articular chondrocytes, where PHEX expression is maximal. Thus, the functional role of SOX9 in regulating PHEX in vivo remains to be better clarified.

Longitudinal bone growth is tightly governed by a complex network of endocrine and paracrine growth factors such as FGFs, thus contributing to different manifestations of the disease at the growth plate. However, localized fluctuations in tissue non-specific alkaline phosphatase (TNAP), pyrophosphate and direct effects of FGF23 are known to play a key role in disease severity and clinical manifestations [3].

In a physiological state, FGF23 is mainly synthesized by cortical and trabecular osteocytes and osteoblasts, but some expression is also found in the GP chondrocytes, mostly resting and hypertrophic [11,38]. Osteocyte-derived FGF23 is predominantly expressed after birth and controls bone mineralization and phosphate metabolism by regulating sodium phosphate cotransporters (NaPi-IIa and NaPi-Iic), adjusting phosphate reabsorption in the renal tubule [15,48]. In this regard, there is a clear connection between FGF23 and NaPi-IIa and NaPi-IIc phosphate channels, as it has been shown that direct administration of recombinant FGF23 reduces the renal expression of NaPi-IIa in Hyp mice and in patients with XLH, while renal expression of NaPi-IIa and/or NaPi-IIc is downregulated in mice with high levels of FGF23 [6,12,16,49]. Transgenic mice overexpressing human FGF23 are smaller and have shorter extremities than wild-type mice [50]. XLH is associated with high circulating levels of FGF23 and extracellular matrix phosphoglycoprotein (MEPE), especially in the bone marrow and the brain. MEPE is a mineralization inhibitor bound by PHEX that appears to be localized in the hypertrophic zone of the GP, as a result of PHEX/PHEX mutation [51,52,53]. While transgenic mice overexpressing FGF23 have phenotypes similar to the clinical characteristics of XLH, mice with deletions in FGF23 show hyperphosphatemia, ectopic mineralization and poorly formed skeletons with extremely low PTH levels; they also show increased 1,25(OH)_2_ vitamin D3, which suppresses PTH levels attenuating an increase in circulating FGF23 [38]. Conversely, parathyroid hormone-related protein (PTHrP), expressed in the resting zone of the GP that promotes chondrocyte proliferation, appears to be increased in XLH [37,54,55].

The FGFs are a family of proteins known to be implicated in the regulation of GP chondrogenesis. Michigami et al. (2019) reported that the expression of FGFR1 and FGFR3, receptors for FGF1 and FGF2, were upregulated in Hyp mice, as were FGFR2 and FGFR4. Furthermore, it is interesting to note that mice overexpressing FGF2, FGF9 or FGF18, similarly to Hyp mice, exhibit suppressed GP chondrocyte hypertrophy, shortened long bones, and dwarfism [12,14,56,57]. Additionally, FGF21 has been shown to be expressed in the perichondrium surrounding the rat postnatal GP, and seems to act as a negative regulator of GP chondrogenesis in humans and animals [57]. In this regard, there is evidence that transgenic mice overexpressing FGF21 are significantly smaller, have reduced tibial length, and display reduced hepatic GH insensitivity when compared with wild-type mice [57]. Moreover, the expression of the WNT signaling components that regulate the differentiation of early chondrocytes was found to be increased in Hyp mice and, in contrast to other publications, FGF23 phosphaturic activity may act in a WNT-independent manner [58].

Interestingly, Na–K–Cl cotransporter 1 (NKCC1), IGF1 and aquaporin 1 (AQP1), proteins, which are likely involved in chondrocyte enlargement during the hypertophic process, were found to be diminished in Hyp mice [5,29]. Similarly, osteocalcin (OCN) and sialoprotein (BSP) were found to be significantly decreased in hypertrophic chondrocytes while osteonectin (ON) levels were increased. Moreover, the apoptotic activity of hypertrophic chondrocytes and the expression of metalloprotease 9 (MMP9), a key regulator of GP angiogenesis, were found to be reduced in Hyp mice [16,46]. Some authors hypothesize that these results could indicate that MMP9 may lead to delayed chondrocyte apoptosis and cartilage matrix degradation, contributing to GP thickening and increasing cartilage and bone volumes in the primary spongiosa. The vascular endothelial growth factor (VEGF) involved in the expansion of the hypertrophic layer, was also found to be downregulated, suggesting a detrimental vascular invasion [5], while it was also reported that a significant increase in both metalloprotease 13 (MMP13) and tartrate-resistant acid phosphatase (TRAP) in Hyp mice, as well as CLCN7 chloride channel, the ATPase H + lysosomal transporter and the NHEDC2 sodium and hydrogen transporter, traditionally expressed by osteoclasts to acidify the extracellular environment during bone resorption [4,53].

Children with XLH show an over-activation of both extracellular signal-regulated kinase (ERK) and pERK1/2, leading to an expansion of the hypertrophic chondrocyte layer and a decrease in type I collagen in vitro, as well as an upregulation of the mitogen-activated protein kinase (MAPK) signaling pathway in the GP, while serum calcium levels remain normal [5,6,29,59]. In this regard, pERK1/2 inhibition activity in Hyp mice has been related to a partial recovery of cartilage deformities and skeletal abnormalities [5]. In addition, studies using in vitro analyses of primary murine chondrocytes have demonstrated that phosphate mediates hypertrophic chondrocyte apoptosis by activating the caspase-9-dependent mitochondrial pathway [6,16].

XLH is also associated with upregulated 24-hydroxylase in Hyp mice compared with controls, indicating that catabolism of calcitriol is probably increased. Inhibition of bone-derived calcitriol may contribute to rickets by inhibiting chondrocyte differentiation [12,16,36,60,61]. VDR-null mice (hereditary vitamin D-resistant rickets) was demonstrated to have a rachitic GP phenotype, mainly due to an expansion of the late hypertrophic chondrocyte layer, a consequence of impaired apoptosis of these cells as it occurs in hereditary hypophosphatemia [16]. Previous studies showed that diets high in phosphorus, calcium or lactose prevent the development of hyperparathyroidism, rickets and osteomalacia in these mice, suggesting that hypocalcemia, hypophosphatemia or hyperparathyroidism could partly cause rickets.

Aberrant distribution of osteoclasts and evident reduction in the alkaline phosphatase (ALP) signaling in the primary spongiosa, a marker of chondrocyte maturation, could explain the lack of mineralization in bones and, consequently, the profound disorganization of trabeculae in the primary spongiosa typically found in XLH, while ALP activity in the GP is increased [5,62]. Some studies have shown that Hyp mice have abnormal renal prostaglandin production, as FGF23-mediated upregulation of prostaglandin E2 (PGE2) via inhibition of proximal tubule phosphate transport may also contribute to hypophosphatemia; this leads to defective mineralization of bones by increasing calcitriol concentration in both animals and humans [11]. In this regard, a novel mechanism contributing to the mineralization defect in Hyp mice has been described, since it appears that increased local FGF23 production is only partially responsible for impaired mineralization in Hyp mice. Furthermore, TNAP, the expression of which is decreased in Hyp-derived osteocyte-like cells ex vivo and in vitro, leads to the accumulation of PPi, a potent inhibitor of mineralization. Thus, Beck-Nielsen et al. showed that blocking the increase in FGF23–FGFR3 signaling in Hyp-derived osteocyte-like cells, lead to an improved TNAP activity and phosphate production, while decreasing the PPi concentration in vitro [12]. Moreover, Col10a1 expression, which has a functional role in bone repair and remodeling in the physiological state, is upregulated in Hyp osteoblasts and bones, thereby raising the possibility that ectopic production of type X collagen could contribute to the impaired mineralization of the Hyp bone matrix [38,63].

The most important factors related to impaired dynamics of the GP during XLH are summarized in Table 3.

## 5. Treatment Effects in XLH

The treatment of metabolic skeletal dysplasia has been primarily focused on the symptomatic correction of abnormalities in the phosphate–calcium metabolism and/or the surgical correction of skeletal deformities. Therefore, conventional therapy for XLH has been mainly based on the administration of daily doses of oral phosphate and vitamin D metabolites or analogues (commonly alfa-calcidol); this is because phosphate insufficiency and inappropriately low levels of calcitriol [1.25(OH)2D^3^ or active vitamin D] contribute to many symptoms of XLH [16,17,64,65]. Other strategies based on a recombinant growth hormone or therapies targeting FGF23 have shown promising results [29]. Owing to the complexity and multisystemic nature of the disease, patients should be seen regularly by multidisciplinary teams organized by metabolic-bone experts. There is also a big necessity to develop clear recommendations for the diagnosis and management of patients, especially for XLH children, which will help to improve knowledge and guidance for diagnosis and multidisciplinary care, the last review having been conducted in May 2019 [66].

### 5.1. Therapeutic Effects of 1,25D and Pi

Conventional therapy for XLH consists of orally administered phosphate at an initial dose of 20–40 mg/kg/day up to 80–100 mg/kg/day, in 3–5 equidistant fractional doses, plus calcitriol supplements (20–50 ng/kg/day) in 2–3 doses [67]. Liu et al. (2016) stated that daily doses of 1,25D dramatically improved GP maturation normalizing the columnar organization of chondrocytes [47]; they also normalized vertebral height, body weight, bone volume and biomechanical parameters such as FAM20C (which is required for FGF23 phosphorylation), SOST, ANK and PHOSPHO1 [6]. Even though this treatment strategy improves hormonal homeostasis and ionic mineralization, it causes secondary complications such as hypercalcemia, hypercalciuria, digestive complications, secondary and/or tertiary hyperparathyroidism, ectopic calcifications and nephrocalcinosis associated with active vitamin D dosage [6,12,67,68]. Furthermore, the lack of adherence seems to be one of the main problems of this treatment, especially in adolescence [65,66].

There is a huge interest in identifying novel therapeutic treatments that reduce the need for oral phosphate. Studies in the mouse model of XLH demonstrated that daily 1,25D treatment without phosphate supplementation improved both serum phosphate levels and skeletal phenotype, but increased serum and bone FGF23 levels in mice [69]. In this regard, the overexpression of FGF23 has also been associated with various aspects of cardiovascular disease; recent studies have indicated that high circulating FGF23 levels lead to the development of left ventricular hypertrophy, although reports of cardiovascular abnormalities and hypertension in patients with XLH are rare and inconsistent, and authors determined that multiple factors presumably played a role [12].

In addition, it has been demonstrated that Eldecalcitol ED71 (ED71 or [1α,25-dihydroxy-2β-(3-hydroxypropyloxy) vitamin D3], an active vitamin D3 analog approved for osteoporosis therapy in Japan, is a more active stimulator of calcium or Pi absorption in normal rat and mouse intestines compared with 1,25D; however it causes FGF23 resistance in Hyp mice similar to conventional treatment [68]. Although the effect of ED71 on Pi metabolism and its skeletal effects for hypophosphatemic rickets are not fully understood, Kaneko et al. (2018) demonstrated, in Hyp homolog XLH, a rescued GP cartilage and an improved calcification in both the hypertrophic zone of the GP and the metaphyseal trabeculae, which contributes to bone strength [68]. However, gene expression and circulating FGF23 levels were markedly increased, and the expression of genes encoding ALP and PTH were attenuated. Importantly, the age at which treatment starts influences the trajectory of growth. In this regard, adult height was higher in those patients treated with phosphate and 1.25(OH)2D supplements in the first year of life than those starting treatment after one year at age 4.

### 5.2. Therapeutic Effects of GH

The use of GH in patients with XLH leads to a significant increase in linear growth and height, as well as a normalization of PTH levels, an increase in calcitriol levels, and a reduction in renal phosphate excretion. However, the main complications of this treatment are skeletal deformities and body disproportion due to the failure in correcting the GP abnormalities, as it might implicate a higher growth in some zones compared to others [3,66]. In this regard, several studies have shown that, indeed, GH promotes proliferation and maturation of chondrocytes, as well as an increment in the expression of proteins implicated in chondrocyte hypertrophy; in turn, this increases the growth rate, but does not normalize the GP structure in patients with XLH [4]. Interestingly, GH treatment, despite having a positive effect on final length and bone quality, was not able to improve the aberrant GP structure in Hyp mice. Thus, the use of GH might potentially increment the risk of bone deformities in pediatric patients with XLH. On the other hand, it has been shown that co-administration of GH, together with a MAPK inhibitor, not only favors bone mineralization and growth, but also normalizes plaque structure, making it a promising treatment for the disease [5,70].

Maintained treatment with supplements of phosphate and 1-OH hydroxylated vitamin D metabolites in XLH children accelerates longitudinal growth velocity, but does not fully correct the final height reduction, even when the osseous lesions of active rickets are healed. Consistent alterations in growth hormone (GH) secretion are not present in patients with XLH.

### 5.3. Therapeutic Effects of FGF23-Neutralizing Antibody (FGF23Ab) or FGF23 Signaling Blockade

The standard treatments for XLH patients are based on prescribing oral phosphate and calcitriol. However, it does not completely rescue the rickets, nor does it alleviate all of the associated symptoms. Serious side effects such as nephrocalcinosis and hyperparathyroidism have also been observed. Novel therapies inhibiting the excessive activity of FGF23 or replacement therapy with recombinant FGF23 may be beneficial for those patients with abnormal GP development and several metabolic bone diseases, including XLH [36]. Kaneko et al. (2018) showed that inhibition of FGF23 by different strategies such as pan-inhibition of FGF receptors, inhibition of the pathway, and neutralization of FGF23, resulted in an improvement of both hypophosphatemia and bone integrity [68]. Furthermore, administration of FGF23-pathway inhibitors resulted not only in the acceleration of growth and improvement of rickets, but Beck-Nielsen et al. (2010) also demonstrated that enthesopathies improved in XLH patients treated with FGF23-blocking [12]. Overexpressed FGF23 regulates Wnt/β-Catenin Signaling-Mediated Osteoarthritis (OA). Canonical Wnt-ERK inhibitor treatment ameliorates OA abnormalities in subchondral bone and reduces degradative/hypertrophic chondrogenic marker expression in mice [71,72]. A Bianchi et al. demonstrate that FGF23 participates in the osteoarthritic (OA)-induced phenotype switch of chondrocytes (FGF23 was overexpressed in OA chondrocytes) [72]. Can FGF23 inhibitors improve OA during all stages and be beneficial in XLH patients who develop early OA? What is the role of PHEX and FGF23 during the cartilage degradation? These questions are as yet unresolved.

Treatment with a FGF23-neutralizing antibody (FGFG23Ab) from four to eight weeks normalized the femoral ash weight, GP thickness and serum Pi levels as well as increasing circulating 1.25(OH)2D_3_ levels and normalizing growth plate thickness [68]. This direct blockade of FGF23 action has been reported to be more robust and long-lasting than those attained by conventional treatments. Moreover, the use of increasing monthly subcutaneous doses of Burosumab, a human IgG1 monoclonal antibody directed at FGF23, in trials carried out in XLH patients, demonstrated a short-term beneficial effect by slight improvements of linear growth; it also decreased the severity of rickets in children, and osteomalacia related fractures and bone pain in adults. Burosumab also improved renal phosphate reabsorption threshold, as well as serum phosphate and calcitriol levels. These results suggest that Burosumab accelerates growth velocity and improves histomorphometric parameters without aggravating body disproportion unlike conventional therapy [29,64,73]. Burosumab expands the therapeutic horizon, especially for severely affected children and patients with an insufficient response to conventional treatment. However, no clinical studies analyzing the effects of long-term treatment and the impact on adult height are available to date [62]. On the other hand, antagonizing FGF23 activity with directed antibodies (Burosumab treatment) requires at least a monthly infusion, is very costly, and does not alleviate symptoms in many patients, but not all. Hence, it is of the utmost necessity to identify more affordable strategies for therapy.

Dr. Santos’ Laboratory demonstrated that inhibiting the MAPK pathway (FGF23 downstream pathway) in PHEX mice partially rescues growth impairment by normalizing the GP structure, specifically in the hypertrophy zone [5]. The C-terminal FGF23 fragment has also been reported to antagonize FGF23 activity [57,74] emerging as some more affordable candidate therapies to antagonize FGF23 activity in FGF23-mediated hypophosphatemia. However, the effect of two differently modified C-terminal FGF23 fragments on growth and growth plate is still not fully explored.

## 6. Conclusions

X-linked hypophosphatemia is a skeletal disorder characterized by several skeletal features, the main ones being growth retardation, osteomalacia in adults and rickets in children. A better understanding of the alterations in the dynamics and maturation of the GP seems to be the key for improving the growth and bone phenotype. Furthermore, newly developed therapeutic strategies targeting FGF23 have shown promising short-term results in normalizing growth and decreasing the severity of rickets. Nevertheless, further studies analyzing the effect of a long-term treatment and the impact on adult height are needed, and seem to be the next step to further determine the effectiveness of the treatment. The cartilage in general, and the GP in particular, are tissues that are hardly approachable in vivo, and they justify the use of animal models such as the Hyp mouse.

The mechanism by which FGF23 inhibition affects GP and cartilage dynamics needs to be clarified, as well as the predisposition to develop OA, especially in infants; how this is translated into a growth rescue, and whether FGF23 related treatments would be suitable for pediatric patients, also needs clarification. In the same way, it would be attractive to know how the treatments (FGF23 blockers) affect the predisposition to develop OA [58]. Furthermore, the concept of directed medicine has gained special relevance due to the growing need to develop personalized strategies for diagnosis, treatment and monitoring. However, the GP is a tissue with a diverse cellular composition. This cellular variability is reflected in its molecular composition, as well as with the transcriptomic profile. Given the heterogeneity and genetic complexity of this tissue, there is great variability in the response to treatment, depending on the altered molecular pathway. Consequently, gaining a better understanding of GP development and cartilage differentiation might not only promote the development of new therapeutic strategies, but it will have also a great impact on the conceptual understanding of various biological processes that happen during the process of cartilage maturation and differentiation. A better understanding of growth plate dynamics will, therefore, be easily transferable, and can be applied to other diseases with high socio-sanitary impact, such as joint osteoarthritis or chronic kidney failure, where chondrocyte hypertrophy and differentiation also play a relevant role.

## Figures and Tables

**Figure 1 ijms-23-00934-f001:**
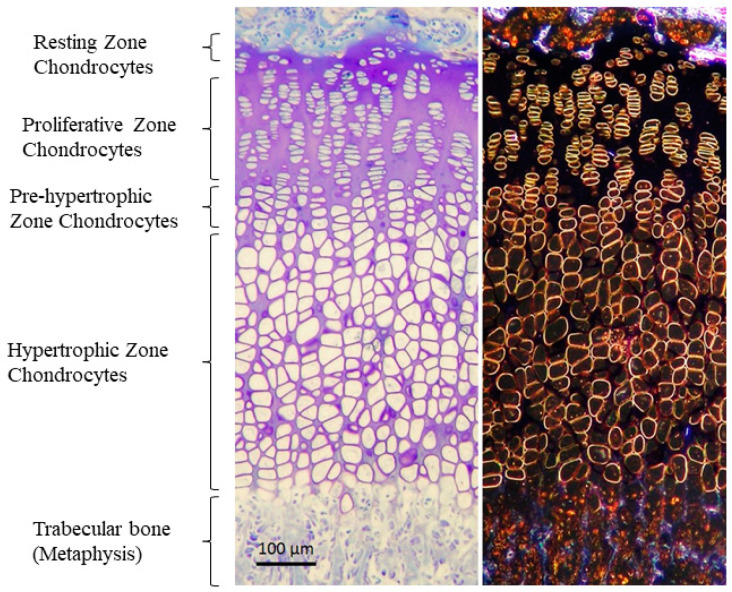
Structure of the GP stained with Toluidine blue in bright field (**left**) and phase contrast (**right**). Four zones are distinguished: (1) germinal or resting zone, which serves as a reservoir for cells and nutrients; (2) proliferative zone or zone of active growth, where chondrocytes proliferate and divide, forming columns oriented in the direction of growth; (3) hypertrophic zone, which is the second zone of active growth, in this case not due to proliferation but due to an increase in the size of cells. Chondrocytes here increase in size due to water entry and protein synthesis, and begin to form a matrix that will later be digested by osteoclasts and calcified by osteoblasts; (4) ossification zone, where the hypertrophic chondrocytes that have reached their maximum size die, leaving a space or lagoon that will be calcified by the osteoblasts, or they can also differentiate into osteoblast. Vascular invasion is also critical for the correct arrival of nutrients, factors, osteoblasts and osteoclasts to the calcification area. Micrographs from the author’s laboratory.

**Figure 2 ijms-23-00934-f002:**
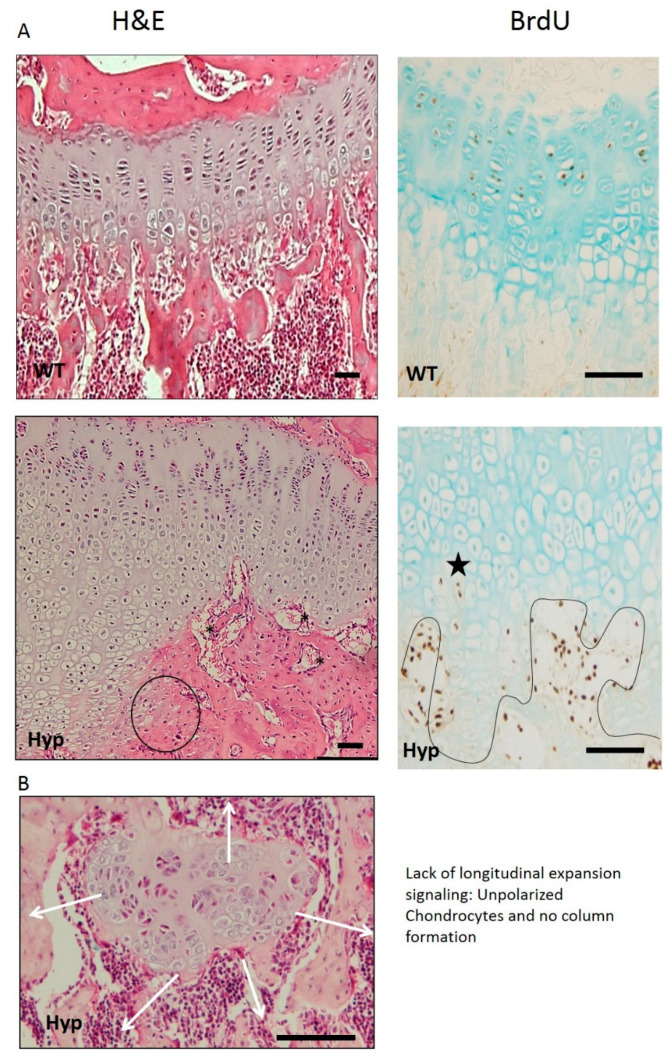
(**A**) Gross appearance and proliferation of the GP. Hyp mice have a very evident alteration of the hypertrophic zone, losing the characteristic columnar pattern. In addition, most aberrant areas appear with flattened chondrocytes, losing the hypertrophic phenotype (black circle). Vascular invasion is clearly affected with vessels without parallel orientation to the columns (black asterisk). BrdU immunostaining shows some hypertrophic chondrocytes returned to a proliferative phenotype (black star) inverting the differentiation pattern. (**B**) Lost of columnar patter of GP chondrocytes. The obvious loss of polarity in some areas of the GP is very remarkable (H&E, 1B). This is very important, since polarity is a critical aspect for the correct organization of the epiphyseal plate and growth in the longitudinal direction. Micrographs from the author’s laboratory.

**Table 1 ijms-23-00934-t001:** Clinical and biochemical characteristics of X-linked hypophosphatemia.

**Skeletal manifestations**
Short stature
Rickets in children
Osteomalacia in adults (less severe in females)
Osteoarthritis (common in the ankles, wrists, knees, feet, and sacroiliac joints)
Joint and bone pain (in adults)
Impaired mobility
Bowed legs (valgus or varus deformities)
Enthesophaty or calcification of tendons, ligaments, and joint capsules (in adults)
Premature cranial synostosis and increased antero-posterior head length
Dental abnormalities (abscesses, enamel and dentin defects) such as oral pain, delayed eruption, enlarged pulp chambers, and taurodontism of permanent molars.
Spinal cord stenosis
Hearing loss (in adults)
**Renal disorders**
Impaired renal tubular reabsorption of phosphate
Renal phosphate wasting
**Biochemical abnormalities**
Hypophosphatemia
Elevated circulating FGF23 concentrations
Low or normal levels of 1.25(OH)_2_D or calcitriol
Normal or slightly increased levels of serum PTH
Increased levels of circulating α-Klotho
Elevated levels of serum alkaline phosphatase
Normal calcemia
FGF23: Fibroblast growth factor 23; 1.25(OH)_2_D: 1.25-dihydroxyvitamin D; PTH: Parathyroid hormone.

**Table 2 ijms-23-00934-t002:** Hypophosphatemic syndromes of genetic origin associated with elevated levels of fibroblast growth factor 23 (FGF23).

*Syndromes*	*Cause*
XLH	*PHEX* *mutations*
ADHR	*FGF23* *mutations*
ARHR1	*DMP1* *mutations*
ARHR2	*ENPP1* *mutations*
ARHR3 or Raine syndrome	*FAM20C* *mutations*
Osteoglophonic dysplasia *	*FGFR1* *mutations*
Jansen-type metaphyseal chondrodysplasia	*PTHRP1* *mutations*
Epidermal nevus syndrome	*FGFR3* *mutations*
DAKGM	*KL* *mutations*
McCune–Albright syndrome	*GNAS*
Epidermal nevus syndrome	

XLH, X-linked hypophosphatemia; ADHR, autosomal-dominant hypophosphatemic rickets; ARHR, autosomal recessive hypophosphatemic rickets. PHEX, phosphate-regulating endopeptidase homolog X-linked; FGF23, Fibroblast growth factor 23; DMP1, Dentin Matrix Acidic Phosphoprotein 1; ENPP1, Ectonucleotide Pyrophosphatase/Phosphodiesterase 1; FAM20C, FAM20C Golgi-Associated Secretory Pathway Kinase; FGFR1, fibroblast growth factor receptor 1; PTHRP1, Parathyroid Hormone-Related Protein; DAKGM, Diseases Associated with Klotho Gene Mutations; KL, Klotho gene; GNAS, GNAS Complex Locus. * Some patients have elevated abnormal unmineralized bone zones producing high circulating FGF23 levels.

**Table 3 ijms-23-00934-t003:** Molecular factors implicated in the pathogenesis of the XLH growth plate phenotype.

Regulators		Expression/Effect at the GP	X-Linked Hypophosphatemia
Systemic regulators			
	GH	Resting and Proliferative zones	Normal/Low levels
	Serum Calcium	Hypertrophic	Normal levels
	Urine calcium	−	Low levels
	Circulating phosphate	−	Increased
	Urine phosphate	−	Increased
	1.25(OH)_2_D_3_	Proliferative and hypertrophic	Normal/Low levels
	PTH	Hypertrophic	Normal/high levels
Local regulators			
	FGF23	Prehypertrophic	Increased
	FGFR1	Prehypertrophic and hypertrophic	Increased
	FGFR3	Prehypertrophic and hypertrophic	Increased
	PTHrP	Prehypertrophic	Increased levels
	MEPE	Hypertrophic	Increased levels
	AQP1	Hypertrophic	Decreased levels
	NKCC1	Hypertrophic	Decreased levels
	Col10a1	Hypertrophic	Increased expression in bone
	MMP13	Hypertrophic	Increased levels
	MMP9	Hypertrophic	Decreased levels
	VEGF	Hypertrophic	Decreased levels
	IGF1	Hypertrophic	Decreased levels
	pERK1/2	Proliferative and prehypertrophic	Increased levels
	ALP	Hypertrophic	Increased levels
	OCN	Hypertrophic	Decreased levels
	BSP	Hypertrophic	Decreased levels
	ON	Hypertrophic	Increased levels
	TRAP	Late hypertrophic	Increased levels
	CLCN7	Hypertrophic	Increased levels
	ATPase H^+^	−	Increased levels
	NHEDC2	−	Increased levels

GH, Growth Hormone; 1.25 (OH) 2D3, active metabolite of vitamin D; PTH, Parathyroid Hormone; FGF23, Fibroblast Growth Factor 23; FGFR 1/3, Fibroblast Growth Factor Receptor 1/3; PTHrP, Parathyroid Hormone related Protein; MEPE, Extracellular Matrix Phosphoglycoprotein; AQP1, Aquaporin 1; NKCC1, Na K Cl cotransporter; Col10a1, Type X collagen; MMP13/9, Matrix Metalloproteinase 13/9; VEGF, Vascular Endothelial Growth Factor; IGF1, Insuline-like Growth Factor 1; pERK, phospho Extracellular signal-Regulated Kinase; ALP, Alkaline Phosphatase; OCN, Osteocalcin; BSP, sialoprotein; ON, osteonectin; TRAP, Tartrate-resistant acid phosphatase; CLCN7, Chloride Voltage-Gated Channel 7; NHEDC2, Na+/H+ exchanger-like domain-containing protein 2.

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
