# Peer review of "Cellular and Molecular Alterations Underlying Abnormal Bone Growth in X-Linked Hypophosphatemia"

_ijms, 2022, doi:10.3390/ijms23020934_

Round 1
Reviewer 1 Report
This is an excellent written review for which i have only some minor comments to the authors
- in table 2: I believe that the addition of a column describing the gene involved will make the table more valuable.
- in table 2: since in the latest consensus statement on clinical practice recommendations for the diagnosis and management of XLH (Haffner et al. Nature reviews of Nephrologyvol 15 2019) hypoposphatasia associated with Rayne syndrome is also acro-named ARHR3
- in table 2: i had the impression that osteoglophonic dysplasia is characterised by normal FGF23 levels
- in table 2: does diseases associated with Klotho gene mutations should be in the table
in page 9 line 337: why TRAP is referred as acidic tartarate resistant acid phosphatase? Just "tartarate resistant acid phosphatase" is enough.
in table 3: the second column is headlined as protein however several molecules below are not proteins Ie. serum calcium.
in the section 5 Treatment effects in XLH: i have not noticed any reference to the recently published consensus clinical practice recommendations endorsed by several international societies.
Reviewer 2 Report
Fuente and colleagues provide a thorough overview of the cellular and molecular alterations underlying abnormal bone growth in X-Linked Hypophosphatemia. The text is lengthy in some respect and could be shortened while remaining exhaustive (for example, with respect to the basics of GP; also, the parts related to OA and cartilage can be deleted). Moreover, we recommend revising the English text since there are a number of typo. More importantly, the authors should clarify the source of the pictures provided in Figures 1 and 2 (not clear at present), and specify whether permission to use has been obtained.
Reviewer 3 Report
Fuente et al provide a review on XLH focusing on growth plate in Hyp mice.
Major points: The authors quote Yang L et al [38] and attempted to argue that “a chondrocyte-to-osteoblast lineage” might underlie pathological basis in XLH without further explanations. How could the trans-differentiation affect XLH? What should be explored in the future in terms of GP? Why are the authors preoccupied by longitudinal growth? The authors should provide explanation?
Minor points:
Line 45. “The bone growth is always the results of the GP activity” sounds incorrect considering periosteal growth of cortical bone. Please remove “always”.
Line 49. “Disturbances of the skeletal growth are always associated with alterations in the activity/regulation of the GP cartilage” also sounds incorrect. Please remove “always” from the sentence.
Line 62. Please verify www.PHEXdb.mcgill.ca/. I could not visit the site.
Table 2. “ADHR, dominant autosomal…” should be “ADHR, autosomal dominant…” and “ARHR, recessive autosomal…” should be ““ARHR, autosomal recessive…”.
Line 111. “PHEX that begins in exon 16…” should be “the PHEX gene that begins in exon 16…”
Line 120 and throughout the manuscript. “2” in 1,25(OH)2D should be subscript.
Line 129. “total width in the direction of growth” may be better than just “total width”.
Line 138. Something is wrong with “develop Osteoartritis (OA) eary [27,28]”.
Line 147. “XLH is a cartilage disorder” sounds too abrupt. “can be considered as a cartilage disorder”? This is important because the authors describe “little is known about the underlying GP alterations in this disease” (Line 208).
Line 273.Lui et al (2916) needs ref. number.
